# Predictors of Employment Status for Persons with Bipolar Disorder

**DOI:** 10.3390/ijerph19063512

**Published:** 2022-03-16

**Authors:** Shu-Jen Lu, Tsan-Hon Liou, Ming-Been Lee, Chia-Feng Yen, Yen-Ling Chen, Reuben Escorpizo, Ay-Woan Pan

**Affiliations:** 1Vocational Rehabilitation Resource Center for Individuals with Disabilities, Hsinchu 31064, Taiwan; sjl470924@yahoo.com.tw; 2Department of Psychiatry, Taipei Tzu Chi Hospital, New Taipei City 231405, Taiwan; 3Department of Physical Medicine and Rehabilitation, Shuang Ho Hospital, Taipei Medical University, New Taipei City 235041, Taiwan; peter_liou@s.tmu.edu.tw; 4Department of Physical Medicine and Rehabilitation, School of Medicine, College of Medicine, Taipei Medical University, Taipei 110301, Taiwan; 5Department of Psychiatry, National Taiwan University College of Medicine, Taipei 10617, Taiwan; mingbeen@ntu.edu.tw; 6Department of Psychiatry, Shin Kong Wu Ho-Su Memorial Hospital, Taipei 11101, Taiwan; 7Department of Public Health, Tzu Chi University, Hualien 97004, Taiwan; mapleyeng@gmail.com; 8Institute of Biophotonics, National Yang-Ming University, Taipei 11221, Taiwan; yp2781@gmail.com; 9Department of Rehabilitation and Movement Science, College of Nursing and Health Sciences, University of Vermont, Burlington, VT 05401, USA; escorpizo.reuben@gmail.com; 10Swiss Paraplegic Research, 6207 Nottwil, Switzerland; 11School of Occupational Therapy, College of Medicine, National Taiwan University, Taipei 10051, Taiwan; 12Occupational Therapy Division, Department of Psychiatry, National Taiwan University Hospital, Taipei 100229, Taiwan

**Keywords:** mental illness, ICF, vocational rehabilitation

## Abstract

Bipolar disorder is characterized by manic and depressive episodes and can be a lifetime condition. Bipolar disorder has been found to be associated with various types of disabilities, including low employment rate and high dependence on public aid. The purpose of this study is to identify factors related to being employed for persons with bipolar disorder. Nine thousand eight hundred and eighty-six subjects with bipolar disorder were collected between July of 2012 and November of 2013 and retrieved from Taiwan national disability database on May of 2014. The mean age of the sample is 45.41 (SD = 10.5), with 64% as female. Logistic regression was used to examine the log odds of the predictive variables on outcome of employment. A Receiver Operating Characteristics analysis was applied to locate the cutoff score of World Health Organization Disability Assessment Schedule 2.0 for being employed. All demographic variables were found to be significantly correlated with employment status among subjects. The Receiver Operating Characteristics results revealed that those subjects whose scores were below 33.57 had about a four-fold higher probability of being in employment than those whose scores were above 33.57. The result provides insights into future research effort and intervention design aimed at helping persons with bipolar disorder to obtain gainful employment.

## 1. Introduction

Bipolar disorder is characterized by manic and depressive episodes, with depression being the major mood condition [1,2]. The onset of bipolar disorder typically occurs in the late teens or early adulthood, a period which is widely recognized as being key in the development of academic, occupational and social skills [3]. 

Bipolar disorder has been found to be associated with various types of disabilities, including increased suicidal behavior, increased usage of healthcare resources (and the associated costs), higher unemployment, higher dependence on public assistance, lower annual income, increased work absenteeism directly attributable to illness, a reduction in work productivity, poorer overall functioning, lower quality of life and a general reduction in life expectancy [1,4,5]. Bipolar disorder is also found to be associated with lower medication adherence rates, lower symptomatic recovery rates and lower functional outcomes [6].

In terms of the global burden of diseases, bipolar disorder is ranked as the 22nd highest [7], with the World Health Organization (WHO) identifying bipolar disorder as being among the top ten causes of years lost to disability [8]. Supported living arrangements and occupational problems contribute to the estimated US$45 billion loss in productivity in the US, whilst the annual cost of bipolar disorder itself is estimated at 31.4 billion [9]. 

The study results of Kessler et al. revealed that bipolar disorder accounts for annual losses of 65.5 working days per worker, whilst the annual losses for major depression are around 27.5 days [10]. When work becomes a serious issue in individuals with all types of impairment, it is very important to understand those factors related to employment [11]. In general, work capacity in persons with bipolar disorder is affected by the impacts of various clinical and demographic factors [12]. These predictors can be divided into six categories, comprising of cognitive performance, symptomatology, sociodemographic factors, course of illness, clinical variables and other personal factors [13]. Among these factors, cognitive impairment had moderate and positive effects on employment and global functions, such as verbal memory performance, executive function, years of education and attention [14,15]. These cognitive abilities are closely related to learning new tasks, acting purposefully and making decisions, all of which are skills that are likely to be prerequisites to success in most types of employment [16]. 

In symptomatology, among all high-risk groups, people with depression had the longest periods of sick leave [17], with an increase in the severity of depressive symptoms being associated with a gradual increase in dysfunction, essentially as a result of the impact of the disease leading to reduced participation in work [18,19]. Furthermore, educational attainment is known to be a predictor of rates of employment and work functioning, with such education having direct impacts on occupational status in the bipolar disorder population [16]. Other sociodemographic factors, such as age, gender, migration and residence, are also found to affect the work performance and employment rates of these patients [18]. 

However, it should be noted that these factors can give rise to diverse results based upon time, country or culture; indeed, the findings of the prior related studies indicate that focusing on one single factor affecting the employment of persons with bipolar disorder is inappropriate, as both physical and mental factors may come into play, and these can be very difficult to identify. Thus, it is necessary to create a framework within which researchers and clinicians can account for these multiple factors; and indeed, the WHO does provide such a framework in the World Health Organization Disability Assessment Schedule 2.0, generally referred to as WHODAS 2.0, for the evaluation of functioning, disability and health. 

There are six domains of the WHODAS 2.0, comprising of cognition, mobility, self-care, getting along, life activities and participation [20]. Following the use of the WHODAS 2.0 in some studies as a method of measuring functional impairment [21], the task force of the Diagnostic and Statistical Manual of Mental Disorders, Fifth Edition (DSM-5)—the principal authority for psychiatric diagnoses – considered the WHODAS 2.0 to be a better assessment of disability for clinicians and researchers than the ‘Global Assessment of Functioning’ (GAF) scale.

In their attempts to develop a standardized assessment for people with disabilities, and to provide eligibility criteria for the consideration of subsidy [22,23], in 2012, Taiwanese government published an amendment to the Persons with Disabilities Rights Protection Act, which authorized the formation of a Taiwanese ICF taskforce, leading to this taskforce developing the WHODAS 2.0 traditional Chinese version [24]. Based upon this process, individuals with disabilities relating to mental illness are now regarded as being far more numerous than those with other types of chronic health conditions [25,26]. 

Our primary research aims in the present study are addressed by using the ICF framework and the WHODAS 2.0 scores as the means of gaining a better understanding of the key factors affecting the individual employment status of persons with bipolar disorder in Taiwan. Our main research questions are:(1)Are there significant differences of the demographics between employed and unemployed persons with bipolar disorder?(2)Are there significant differences of the domain scores and summary score of the WHODAS 2.0 between employed and unemployed persons with bipolar disorder?(3)What are those predictive factors related to work status for persons with bipolar disorder?

## 2. Materials and Methods

### 2.1. Participants

Of the original sample of 15,465 persons with bipolar disorder adopted for this study, information on only 9886 of the patients that meet our criteria was included. Those samples were collected between July of 2012 and November of 2013 by trained professional members. We access the dataset on May of 2014. The remaining 5579 samples were excluded due to (i) not within the labor force age range (age < 18 or >65, *n* = 1294), (ii) personal or diagnosis information missing (no employment information, *n* = 563;WHODAS 2.0 incomplete, *n* = 2274), (iii) zero disability levels or disability information missing (*n* = 17), (iv) had comorbidity with other illnesses (*n* = 167), and (v) repeat cases (*n* = 1264). The 9886 participants were subsequently divided into two groups, employed (*n* = 1846) and unemployed (*n* = 8040), based upon their current employment status. The average age of the total sample is 45.41 years old with standard deviation as 10.5; sixty-four percent of the sample are female; the majority of the sample had educational level of senior high school (41.9%); and 50.7% of the sample had mild severity of impairment (Table 1). 

### 2.2. Measures

The full version of the WHODAS 2.0 (36 items) measures average functioning in daily situations for the preceding 30-day period using a five-point scale ranging from 0 (no disability) to 4 (extreme disability) and surveys the six domains of functioning. ‘Cognition’ (6 items), evaluates understanding and communicating ability; ‘Mobility’ (5 items), evaluates physical movement and getting around; ‘Self-care’ (4 items), evaluates the subjects’ hygiene, dressing, eating and staying alone; ‘Getting along’ (5 items), evaluates the subjects’ interaction with other people; ‘Life activities’ (8 items), evaluates their domestic responsibilities, leisure, work, and school; and ‘Participation’ (8 items) evaluates whether the participants are able to join in with community activities. The Chinese version of the WHODAS 2.0 has been shown to be a valid and reliable instrument in Taiwan [23]. 

As compared to the WHOQOL-BREF, the respective reliability of the Cronbach’s *α* and the ICC in the WHODAS 2.0 traditional Chinese version were found to be 0.73–0.99 and 0.80–0.89, whilst the content validity was deemed good (r = 0.70–0.76) and the concurrent validity was regarded as excellent (*p* < 0.5). ‘Exploratory factor analysis’ (EFA) revealed that the construct validity was supported [24].

### 2.3. Statistical Analysis

In an attempt to gain a good understanding of the influences of demographic factors on the employment status of persons with bipolar disorder, we analyzed the means, standard deviations and percentages, and then carried out comparisons between the two employed and unemployed groups, with an independent t-test being applied for age, along with a Chi-square test for other factors, including gender, education level, dwelling (living in the community or an institution), and urbanization level. ‘Receiver operating characteristic’ (ROC) was used to identify an appropriate cut-off point capable of distinguishing the individual employment status of the participants; a ROC curve was generated by plotting the cumulative distribution function of the WHODAS 2.0 scores of the participants and their employment status. The area under the ROC curve estimates the WHODAS 2.0 scores as referral criteria for predicting the probability of returning to work. 

The Pearson’s correlation was used to identify the correlation between WHODAS 2.0 domains. The logistic regression analyses were used to identify predictors of the employment status for persons with bipolar disorder. The analyses were carried out using SPSS 19.0 statistical software.

## 3. Results

A total of 9886 participants with bipolar disorder were included in the final analysis, of which 18.67% (*n* = 1846) were in employment and 81.32% (*n* = 8040) were in unemployment. The participants generally had a greater likelihood of being female (64%), with a mean age of 45.41 years, a senior high school education (*n* = 1731, 41.9%), dwelling within the community (*n* = 9343, 94.7%), a greater likelihood of living in a city (*n* = 3131, 31.7%), core city (*n* = 2255, 22.8%), or boom town (*n* = 2171, 22%), and with only mild impairment (*n* = 5017, 50.7%). Table 1 shows the different profiles of the group of employed participants (*n* = 1846) and those that were unemployed (*n* = 8040) based upon gender, age, education, and place of residence. The age of the unemployed group was found to be significantly greater than that of the employed group (46.03 vs. 42.69; *p* < 0.001) with the percentage of unemployed female participants being significantly higher than that of the male participants (5328 (66.3%) vs. 1002 (54.3%); *p* < 0.001). 

Both the number of participants who had received education at senior high school or above (employed group = 68.8%; unemployed group = 53.0%) and the proportion of the participants who resided within the community (employed group = 99.0%; unemployed group = 93.7%) were found to be significantly higher in the employed group than the unemployed group (*p* < 0.001). The percentage of urbanized participants was also found to be significantly higher in the employed group (24.9%) than in the unemployed group (22.3%) (*p* < 0.05), with those participants with only mild impairment (63.9% in employed group vs. 47.7% in unemployed group) having significantly higher probability of being employed than those with moderate, severe, or profound impairment (*p* < 0.05). The answer of the first research question was that there were significant differences of the demographic variables between two groups. 

Comparisons of the functional scores between the groups of employed and unemployed persons with bipolar disorder across the six domains of the WHODAS 2.0. demonstrated that significant differences in functioning were discernible between the employed and unemployed participants in all domains and summary score (*p* < 0.001) (Table 2). The answer of the second research question was that there were significant differences of the domain scores and summary score of the WHODAS2.0 between two groups.

We examined the correlation of the scores obtained from six domains of the WHODAS 2.0 using Pearson correlation coefficients. As shown in Table 3, domain 1 ‘Cognition’ was found to have the highest correlation with domain 4 ‘Getting along’ (0.693 coefficient), whilst domain 3 ‘Self-care’ exhibited the lowest correlation with domain 4 ‘Getting along’ (0.423 coefficient), although all of the correlation coefficients were found to have two-tailed significance at the 1% level.

The results of the ROC analysis of the summary scores reveal that the cut-off point for the WHODAS 2.0 scores was 33.57, as shown in Figure 1, which by adapting score of 33.57 as the optimal cut-off point, the sensitivity was 70%, whilst specificity was 67%; this may reflect the 73.7% of the participants who were employed versus those who were unemployed (AUC = 0.737). 

Table 4 reveals that those participants with low WHODAS 2.0 summary scores (≤33.57) were more likely to be employed (OR = 4.61; 95% confidence interval [CI]: 4.12–5.16) than those with higher WHODAS 2.0 summary scores (>33.57). A greater likelihood of being in employment was also found for younger age groups (ages 18–45 and 46–55), as compared to those in the 56–64 age group (O = 3.44/3.05; 95% CI: 2.85–4.14/2.50–3.72), whilst a lower likelihood of being in employment was discernible for female participants, as compared to male participants (OR: 0.62, 95% CI: 0.56–0.70). 

With regard to urbanization, residents in the core cities had the greatest likelihood of being actively employed, as compared to the greater likelihood of unemployment among those in the boom towns (OR: 0.798, 95% CI: 0.68–0.94), general towns (OR: 0.71, 95% CI: 0.58–0.86), and rural towns (OR: 0.59, 95% CI: 0.38–0.92). Participants living in institutionalized settings were also found to have a substantially greater likelihood of being unemployed (OR: 0.13, 95% CI: 0.08–0.21) than those living in the community. As compared to mildly impaired participants, a higher likelihood of unemployment was found for participants with moderate (OR: 0.669, 95% CI: 0.595–0.753), severe (OR: 0.466, 95% CI: 0.361–0.602), and profound (OR: 0.388, 95% CI: 0.182–0.828) impairment. Therefore, the answers to the third research question were that the summary score of the WHODAS2.0, age, gender, urbanization, residence, and severity of impairment were predictors of employment status.

The subjects can also be separated based upon the cut-off scores of each domain, as shown in Table 5; those with the lower scores were found to have a higher probability (1.6 to 2.3 times) of being employed.

## 4. Discussion

The findings reported in this study provide additional evidence in support of the results of the prior related studies. Specifically, our results indicate that persons with bipolar disorder with higher scores (severe functional impairment), as evaluated by the WHODAS 2.0 Chinese version in all six domains of cognition, mobility, self-care, getting along, life activities, and participation, had higher rates of unemployment [27]. The uniqueness of our study stems from our use of the ICF framework for the evaluation of the severity of the disability, an approach that has not yet been widely adopted for the examination of employment participation within the related psychiatric literature. However, the lacking information on symptoms and functional impairment remain a problem in evaluation [28]; Hence, the WHODAS 2.0 was used to provide six different sets of information on disability to facilitate an increase in evaluation accuracy, yet it is still not in widespread use; thus, we aim to address this issue in the present study.

We adopted ROC analysis in the present study as the means of carrying out an investigation into the employment status of individual with bipolar disorder based upon different domains of the WHODAS 2.0. The total WHODAS 2.0 scores of our unclassified sample, comprising of a total of 9886 persons with bipolar disorder, were initially analyzed along with their employment status using the ROC curve, with the results revealing a cut-off point of 33.57; those with scores below 33.57 had almost four times the probability of being in employment than those with scores above 33.57. 

Furthermore, all of the demographic variables on our subjects, including age, gender, residence, degree of urbanization, and disability level, were found to be correlated with employment. Consequently, these cut-off points can provide specific values and information from the WHODAS 2.0, which could be of significant value when applied to clinical practice through practice guidance and intervention implementation. Based upon its high discriminative ability (AUC = 0.737), we believe that the WHODAS 2.0 provides a powerful tool for increasing the accuracy of the employment rate of persons with bipolar disorder.

We hypothesize in this study that the employment status of the persons with bipolar disorder is significantly associated with disability; that is to say, people with severe impairments in cognition, mobility, self-care, social interaction, the ability to cope with daily life, and social participation, will also experience significant difficulty finding employment. The advantage of WHODAS 2.0 is that it includes a number of variables that are generally found to be associated with employment status. 

The logistic regression model involved in our analysis indicates that disability is not the only factor affecting the employment of participants, indeed, all of the demographic variables are found to be highly correlated with employment status. The results of several prior related studies have also demonstrated that the employment status of people with mental illness is related to various demographic, environmental, and disease-related variables [29,30,31]. Other individual and/or environmental variables that may also affect employment outcomes, include work history [32], motivation [33], rehabilitation services [34,35], social support [33], and stigma [30]. 

Vocational rehabilitation practitioners should take all of these multiple factors into consideration, in conjunction with our findings in the present study, when offering employment services or designing intervention plans for persons with bipolar disorder. For example, if training courses can place increasing focus on enhancing mobility and improving the ability of the patients to take care of themselves, this may result in raising their probability of gaining employment. Furthermore, WHODAS 2.0 may also be useful as an instrument for screening persons with bipolar disorder for prevocational training.

There are, nevertheless, some limitations of our study that need to be addressed. Firstly, the actual condition of the subjects may not be totally reflected in their self-evaluation reports due to several reasons, such as personal intention. Secondly, the strong-affecting factors, such as education, are not required on the WHODAS2.0 form. Thirdly, the medication status, which might be a key factor in influencing the employment possibility, was not analyzed due to absence of data. Such a variable of medication use is suggested to be included in future research agenda. 

## 5. Conclusions

Based on WHODAS 2.0 scores, persons with bipolar disorder who had scores below 33.57 would have a higher probability of being in employment. According to their demographic information, participants who were younger (18–45 and 46–55 years), male, living in the community or in core cities, had the greatest likelihood of being actively employed, as compared to participants who were older (56–64 years), female, living in an institution or in other types of residence. Our findings indicate there was significant relationship between the employment status and the functional impairment, as measured by WHODAS 2.0, of persons with bipolar disorder. Furthermore, WHODAS 2.0, as a tool, can provide the evaluation of employment status of disability through the six domains, whilst also providing rehabilitation practitioners with a new method of understanding ways of improving their employment services. Finally, government policy can be established using WHODAS 2.0 scores above 33.57 as a reference criterion, leading to those persons with bipolar disorders being provided with more appropriate vocational rehabilitation services. 

## Figures and Tables

**Figure 1 ijerph-19-03512-f001:**
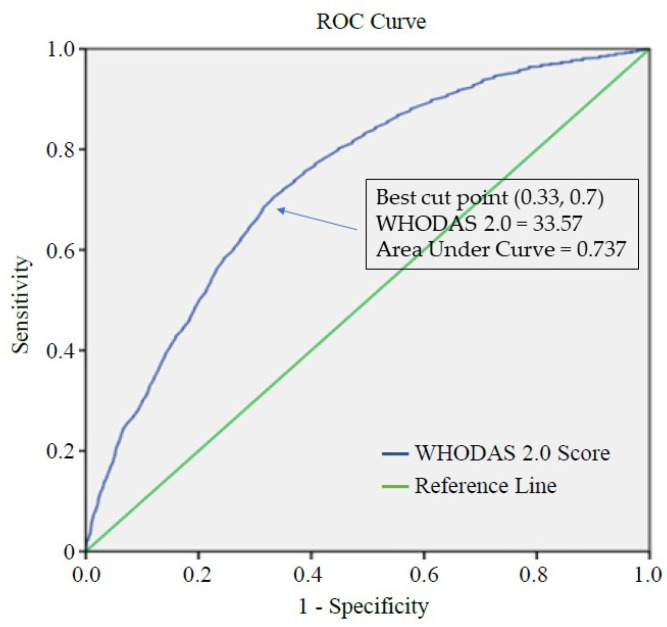
ROC analysis used to determine the optimal cut-off point for the WHODAS 2.0 score. *Note:* WHODAS 2.0 Score = 33.57; Area under Curve = 0.737; Sensitivity = 0.70; 1-Specificity = 0.33. Sensitivity/(1-Specificity) or LR+ = 2.12.

**Table 1 ijerph-19-03512-t001:** Demographic characteristics of the subjects (*n* = 9886).

Variables	Employed ^a^ (*n* = 1846)		Unemployed ^a^ (*n* = 8040)		All (*n* = 9886)		*χ*[2]/*t*-Test *p*-Value
	No.	%	No.	%	No.	%	
Gender							
Male	844	45.7	2712	33.7	3556	36.0	<0.001 *
Female	1002	54.3	5328	66.3	6330	64.0	
Age in years							
Mean (SD)	42.69	(9.61)	46.03	(11.31)	45.41	(10.5)	<0.001 *
Education (*n* = 4128)							
≤Elementary school	55	7.8	717	21.0	772	18.7	<0.001 *
Junior high school	166	23.4	888	26.0	1054	25.5	
Senior high school	353	49.8	1378	40.3	1731	41.9	
≥University	135	19.0	436	12.7	571	13.8	
Residence							
Community dwelling	1824	99.0	7519	93.7	9343	94.7	<0.001 *
Institution	18	1.0	506	6.3	524	5.3	
Urbanization level ^b^							
Core city	460	24.9	1795	22.3	2255	22.8	0.015 *
City	614	33.3	2517	31.3	3131	31.7	
Boom town	388	21.0	1783	22.2	2171	22.0	
Traditional industrial	105	5.7	477	5.9	582	5.9	
General town	211	11.4	1096	13.6	1307	13.2	
Aging town	40	2.2	217	2.7	257	2.6	
Rural town	28	1.5	155	1.9	183	1.8	
Severity of impairment							
Mild	1180	63.9	3837	47.7	5017	50.7	0.015 *
Moderate	578	31.3	3342	41.6	3920	39.7	
Severe	80	4.3	764	9.5	844	8.5	
Profound	8	0.4	97	1.2	105	1.1	

* Significant level <0.05. ^a^ ‘Employed’ includes people who are hired or self-employed, whilst ‘unemployed’ includes volunteers, students, housekeepers, retired people, and those unemployed for health reasons. ^b^ The current definitions of urbanized areas in Taiwan, according to the ‘ROC Statistical Area Standard Classification’ of the Executive Yuan, are as follows: 1. Population of more than 20,000 people. 2. Areas covered by contiguous areas. 3. Two or more neighboring settlements whose populations total more than 20,000. We use the indicators of ‘population density’, ‘educational level’, ‘percentage of population aged ≥ 65 years’, ‘percentage of population aged 15–64’, ‘percentage of industrial employed’ and ‘percentage of employed population’; these are divided into the seven groups of ‘core city’, ‘city’, ‘boom town’, ‘traditional industrial’, ‘general town’, ‘aging town’, and ‘rural town’, with the ‘aging town’ and ‘rural town’ categories then being merged into a single class, to provide a total of six grades.

**Table 2 ijerph-19-03512-t002:** Comparison of the scores of 6 domains and summary score of the WHODAS 2.0 between employment and unemployment groups for subjects with Bipolar Disorder (*n* = 9886).

Variables	Employment			Unemployment			*p* Value *
	Mean	SD	Median	Mean	SD	Median	
Domain 1	31.13	21.73	30.00	41.02	24.57	40.00	<0.001
Domain 2	12.26	18.43	0.00	22.61	25.26	12.50	<0.001
Domain 3	6.30	13.18	0.00	12.86	19.76	0.00	<0.001
Domain 4	38.60	27.21	41.67	49.15	28.37	50.00	<0.001
Domain 5-1	29.89	28.42	30.00	44.09	31.81	40.00	<0.001
Domain 6	39.81	23.15	37.5	47.57	23.97	45.83	<0.001
Summary	28.16	17.92	25.47	44.37	19.18	43.40	<0.001

* Mann–Whitney U-test; Notes: Domain 1: Cognition (understanding and communicating; 6 items), Domain 2: Mobility (getting around; 5 items), Domain 3: Self-care (4 items), Domain 4: Getting along (5 items), Domain 5-1: Life activities (household; 4 items), and Domain 6: Participation (8 items).

**Table 3 ijerph-19-03512-t003:** Pearson correlation matrix of the WHODAS 2.0 domains (*n* = 9886) ^a^.

Domains ^b^	2		3		4		5-1		6	
	Coeff. ^c^	2-Tail Sig.	Coeff. ^c^	2-Tail Sig.	Coeff. ^c^	2-Tail Sig.	Coeff.^c^	2-Tail Sig.	Coeff. ^c^	2-Tail Sig.
1	0.576 **	***	0.485 **	***	0.693 **	***	0.654 **	***	0.690 **	***
2			0.623 **	***	0.491 **	***	0.538 **	***	0.547 **	***
3					0.423 **	***	0.513 **	***	0.432 **	***
4							0.589 **	***	0.690 **	***
5-1									0.592 **	***

^a^ The nonparametric tests used are Mann–Whitney U-tests.^b^ Domain 1: Cognition (understanding and communicating; 6 items), Domain 2: Mobility (getting around; 5 items), Domain 3: Self-care (4 items), Domain 4: Getting along (5 items), Domain 5-1: Life activities (household; 4 items), and Domain 6: Participation (8 items). ^c^ The correlation coefficients have two-tailed significance at the 1% level. ** *p* < 0.01; *** *p* < 0.001.

**Table 4 ijerph-19-03512-t004:** Logistic regression results on the effects of demographic factors on employment status (*n* = 9886) ^a^.

Variables		B	S.E.	Wald-Stat.	df	Adjusted OR	95% CI		*p*-Value
							Lower	Upper	
WHODAS Summary Score ^b^	High (>33.57) (Ref)								
	Low (≤33.57)	0.764	0.029	714.734	1	4.610	4.122	5.157	<0.0001 ***
Age (years)	56~64 (Ref)								
	18~45	0.451	0.042	117.665	1	3.436	2.852	4.141	<0.0001 ***
	46~55	0.332	0.046	52.756	1	3.051	2.504	3.716	<0.0001 ***
Gender	Male (Ref)								
	Female	−0.237	0.028	69.342	1	0.623	0.557	0.696	<0.0001 ***
Urbanization level	Core city (Ref)								
	City	0.175	0.063	7.701	1	0.930	0.803	1.077	0.3330
	Boom town	0.021	0.070	0.088	1	0.798	0.678	0.939	0.0060 **
	Traditional industrial	−0.027	0.110	0.062	1	0.804	0.623	1.037	0.0930
	General town	−0.099	0.084	1.402	1	0.708	0.582	0.860	<0.0001 ***
	Aging town	−0.092	0.163	0.318	1	0.711	0.486	1.039	0.0708
	Rural town	−0.279	0.192	2.140	1	0.591	0.379	0.922	0.0210 *
Residence	Community dwelling (Ref)								
	Institution	−1.028	0.123	69.464	1	0.128	0.079	0.208	<0.0001 ***
Severity of impairment	Mild (Ref)								
	Moderate	0.127	0.108	1.375	1	0.669	0.595	0.753	<0.0001 ***
	Severe	−0.236	0.135	3.037	1	0.466	0.361	0.602	<0.0001 ***
	Profound	−0.419	0.291	2.075	1	0.388	0.182	0.828	0.0140 *

^a^ Employment status: Employed = 1; Unemployed = 0. ^b^ WHODAS Score groups are separated by the best cut-off point on the ROC curve.* *p* < 0.05; ** *p* < 0.01; *** *p* < 0.001.

**Table 5 ijerph-19-03512-t005:** Summary WHODAS 2.0 scores and area under the curve in the six separate domains ^a^.

Variables	Domains						
	1	2	3	4	5-1	6	Summary
25th percentile	20	0	0	25	10	29.1667	28
Median	35	12.5	0	50	40	45.8333	42
75th percentile	55	31.25	20	66.6667	60	62.5	58
Mean	39.1771	20.6801	11.6367	47.1829	41.4404	46.1199	43.2067
cut-off point	37.5	15.625	5	37.5	35	39.583	33.57
OR	0.531	0.431	0.431	0.542	0.489	0.604	0.217
(1/OR)	(1.88)	(2.32)	(2.32)	(1.85)	(2.04)	(1.66)	(4.6)
95% CI							
Lower bound	0.476	0.364	0.346	0.487	0.439	0.544	0.194
Upper bound	0.592	0.510	0.537	0.602	0.544	0.672	0.243

^a^ The results are assessed under the non-parametric assumption, with the null hypothesis being that the true area = 0.5; 95% CI refer to the asymptotic 95% confidence interval.

## Data Availability

No data were reported here.

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
