# Peer review of "Predictors of Employment Status for Persons with Bipolar Disorder"

_ijerph, 2022, doi:10.3390/ijerph19063512_

Round 1

Reviewer 1 Report

IJERPH_1618956 Review

Title: Predictors of Employment Status for Persons with Bipolar Disorder

Overall:  The article is overall very good.  Additional participant information is needed.

Abstract:  Overall the abstract is good and covers the scope of the paper. The abstract should also include the year and time of year that the study occurred as well as some general demographic information.

Introduction:

Line 21:  Bipolar disorder is usually not classified as a disease so please clarify.

Line 24:  Disability should be plural.

Line 37:  Please clarify where “the statistics” originates.  To what statistics are the authors referring?

Materials and Methods:  The participants section needs to include from where the participant information originated and when the researchers accessed the data.  Later, the researchers should specify when the participants filled out forms (if possible) or at least the year(s) in which the participant data was entered into the system.  The participant information section should also include demographic information (if available) regarding sex, age, location, and other information.  If this information was not available, this must be stated clearly.  I see that some of this is stated later in the results section for the population that was ultimately analyzed (at least gender). 

Results:  Overall very clear and thorough. More specific information is needed as to the percentage or number of participants for each category of demographics.  There should be more specifics as to the number of people at the education level mentioned in addition to numbers associated with the living conditions.  While there is a table with more specific numbers, there should be general information here regarding the same types of percentages listed (like used with the gender/sex demographic). 

Discussion:  Overall very good  This is a relatively straightforward study so the discussion is very clean with good focus on limitations. 

Author Response

Reviewer1

Title: Predictors of Employment Status for Persons with Bipolar Disorder

Overall:  The article is overall very good.  Additional participant information is needed.

Thank you.

Abstract:  Overall the abstract is good and covers the scope of the paper. The abstract should also include the year and time of year that the study occurred as well as some general demographic information.

Answer: Thank you. We have added the year and time of the data collection and general demographic information on abstract on page 1 line 7-11, reading as “Nine thousand eight hundred and eighty-six subjects with bipolar disorder were collected between July of 2012 and November of 2013 and retrieved from Taiwan national disability database on May of 2014. The mean age of the sample is 45.41 (SD=10.5) with 64% as female.

Introduction:

Line 21:  Bipolar disorder is usually not classified as a disease so please clarify.

Answer: We have changed it from disease to condition on page 1 line 24, reading as “Bipolar disorder is characterized by manic and depressive episodes, with depression being the major mood condition.”

 Line 24:  Disability should be plural.

Answer: We changed on page 1, line 27-28, reading as “Bipolar disorder has been found to be associated with various types of disabilities,…”

Line 37:  Please clarify where “the statistics” originates.  To what statistics are the authors referring?

Answer: The statistics refer to the results of the cited literature. Thus, we changed it to “The study results of Kessler et al.  revealed that….” As shown on page 1 line 40.

Materials and Methods:  The participants section needs to include from where the participant information originated and when the researchers accessed the data.  Later, the researchers should specify when the participants filled out forms (if possible) or at least the year(s) in which the participant data was entered into the system.  The participant information section should also include demographic information (if available) regarding sex, age, location, and other information.  If this information was not available, this must be stated clearly.  I see that some of this is stated later in the results section for the population that was ultimately analyzed (at least gender).

Answer: We added the information on page 2, line 98 to page 3, line 100. Reading as “Those sample were collected between July of 2012 and November of 2013 by trained professional members. We access the dataset on May of 2014.”

We also added description of demographic information on page 3 line 106-109, reading as “The average age of the total sample is 45.41 years old with standard deviation as 10.5; Sixty-four percent of the sample are female; Most of the sample had educational level of senior high school (41.9%) and 50.7% of the sample had mild severity of impairment (Table1). “

Results:  Overall very clear and thorough. More specific information is needed as to the percentage or number of participants for each category of demographics.  There should be more specifics as to the number of people at the education level mentioned in addition to numbers associated with the living conditions.  While there is a table with more specific numbers, there should be general information here regarding the same types of percentages listed (like used with the gender/sex demographic).

Answer: Thank you. We added quite a few information in the results from page 3, line 147 to page 5, line 181, reading as “A total of 9,886 participants with bipolar disorder were included in the final analysis, of which 18.67% (n=1,846) were in employment and 81.32% (n=8,040) were in unemployment. The participants generally had a greater likelihood of being female (64%), with a mean age of 45.41 years, a senior high school education(n=1,731, 41.9%), dwelling within the community (n=9,343, 94.7%), a greater likelihood of living in a city(n=3,131, 31.7%), core city(n=2,255, 22.8%) or boom town(n=2,171, 22%), and with only mild impairment (n=5,017, 50.7%). Table 1 shows the different profiles of the group of employed participants (n=1,846) and those that were unemployed (n=8,040) based upon gender, age, education and place of residence. The age of the unemployed group was found to be significantly greater than that of the employed group (46.03 vs. 42.69; p<0.001) with the percentage of unemployed female participants being significantly higher than that of the male participants [5,328 (66.3%) vs. 1,002 (54.3%); p<0.001]. Both the number of participants who had received education at senior high school or above (employed group=68.8%; unemployed group=53.0%) and the proportion of the participants who resided within the community (employed group=99.0%; unemployed group=93.7%) were found to be significantly higher in the employed group than the unemployed group (p<0.001). The percentage of urbanized participants was also found to be significantly higher in the employed group (24.9%) than in the unemployed group (22.3%) (p<0.05), with those participants with only mild impairment (63.9% in employed group vs. 47.7% in unemployed group) having significantly higher probability of being employed than those with moderate, severe or profound impairment (p<0.05). The answer of the first research question was that there were significant differences of the demographic variables between two groups. 

Discussion:  Overall very good This is a relatively straightforward study so the discussion is very clean with good focus on limitations.

Answer: Thank you very much.

Reviewer 2 Report

The authors conducted a study aimed at identifying factors associated with being employed in patients with bipolar disorder, using data from the Taiwan national disability database. The topic is of high interest and relevance. The study seems to be well conducted and I have only a few suggestions to improve clarity of presentations of some parts. 

In the abstract, the following expression is not clear and should be rephrased: "... among subjects of either health condition". 

In addition, it is not clear in the abstract to which score they are referring when they mention the cut-off definition, since the WHODAS 2.0 score is not mentioned.

In the statistical analysis section, "The Pearson" should be replaced with "Pearson's correlation".

The description of the analyses at page 3, lines 133 - 136 is not clear and should be rephrased. 

In addition, the authors should specify whether and how they checked assumptions for the conducted test (e.g. tested normality of distribution to choose parametric or non-parametric tests, assumptions to conduct regression and so on).

At page 5, the authors state: "Table 4 reveals that those participants with low WHODAS 2.0 summary scores 195 (≤33.57) were more likely to be employed". However, since the definition of this cut-off is described in a following section, at this point it might not be clear for a reader how this value was chosen. 

Author Response

Reviewer2 

The authors conducted a study aimed at identifying factors associated with being employed in patients with bipolar disorder, using data from the Taiwan national disability database. The topic is of high interest and relevance. The study seems to be well conducted and I have only a few suggestions to improve clarity of presentations of some parts.

Answer: Thank you.

In the abstract, the following expression is not clear and should be rephrased: "... among subjects of either health condition".

Answer: We deleted “of either health condition” as shown on page 1 line 15, reading as “All demographic variables were found to be significantly correlated with employment status among subjects.”

In addition, it is not clear in the abstract to which score they are referring when they mention the cut-off definition, since the WHODAS 2.0 score is not mentioned.

Answer: Thank you for the question. To help the readers to understand which score the text refers, we added information, reading as “A Receiver Operating Characteristics analysis was applied to locate the cutoff score of World Health Organization Disability Assessment Schedule 2.0 for being employed.” As shown on page 1, line 13.

In the statistical analysis section, "The Pearson" should be replaced with "Pearson's correlation".

Answer: We changed it to "Pearson's correlation" as shown on page 3, line 139.

The description of the analyses at page 3, lines 133 - 136 is not clear and should be rephrased.

Answer: Thank you. We changed it to “The logistic regression analyses were used to identify predictors of the employment status for persons with bipolar disorder. The analyses were carried out using SPSS 19.0 statistical software.” As shown on page 3, line 140-143.”

In addition, the authors should specify whether and how they checked assumptions for the conducted test (e.g. tested normality of distribution to choose parametric or non-parametric tests, assumptions to conduct regression and so on).

Answer: Thank you for the question. We applied logistic regression and ROC curve analysis in the study. For these two analyses, they do not require assumptions of normality, linearity, measurement errors etc. (see https://www.statisticssolutions.com/free-resources/directory-of-statistical-analyses/assumptions-of-logistic-regression/). Therefore, we did not test those assumptions.

At page 5, the authors state: "Table 4 reveals that those participants with low WHODAS 2.0 summary scores 195 (≤33.57) were more likely to be employed". However, since the definition of this cut-off is described in a following section, at this point it might not be clear for a reader how this value was chosen. 

Ans: Thank you. We moved the paragraph “The results of the ROC analysis of the summary scores reveal that the cut-off point for the WHODAS 2.0 scores was 33.57, as shown in Figure 1 that by adapting score of 33.57 as the optimal cut-off point, the sensitivity was 70%, whilst specificity was 67%; this may reflect the 73.7% of the participants who were employed versus those who were unemployed (AUC = 0.737).

<Figure 1 is inserted about here>” to the place before description of Table 4 as shown on page 5, line 207 to page 6, line 212.
